# Dual Probes for Positron Emission Tomography (PET) and Fluorescence Imaging (FI) of Cancer

**DOI:** 10.3390/pharmaceutics14030645

**Published:** 2022-03-15

**Authors:** Richard Yuen, Frederick G. West, Frank Wuest

**Affiliations:** 1Department of Chemistry, University of Alberta, Edmonton, AB T6G 2G2, Canada; ryuen1@ualberta.ca (R.Y.); fwest@ualberta.ca (F.G.W.); 2Cancer Research Institute of Northern Alberta, University of Alberta, Edmonton, AB T6G 2E1, Canada; 3Department of Oncology, University of Alberta—Cross Cancer Institute, Edmonton, AB T6G IZ2, Canada

**Keywords:** cancer imaging, dual probes, fluorescence, positron emission tomography

## Abstract

**Simple Summary:**

Being able to detect and image tumors is extremely important for proper diagnosis and treatment. The most sensitive technique, positron emission tomography (PET), is widely applied for such a purpose. Additionally, fluorescence imaging can be used to visually see the margins between healthy and cancerous tissue during surgery. These two techniques can be combined to optimize patient outcomes by ensuring maximum tumor removal. This review will discuss the work that has been done recently to combine these two imaging capabilities into one imaging agent.

**Abstract:**

Dual probes that possess positron emission tomography (PET) and fluorescence imaging (FI) capabilities are precision medicine tools that can be used to improve patient care and outcomes. Detecting tumor lesions using PET, an extremely sensitive technique, coupled with fluorescence-guided surgical resection of said tumor lesions can maximize the removal of cancerous tissue. The development of novel molecular probes is important for targeting different biomarkers as every individual case of cancer has different characteristics. This short review will discuss some aspects of dual PET/FI probes and explore the recently reported examples.

## 1. Introduction

Accurate and precise detection of tumors is essential for diagnosis, staging, and monitoring of treatment response in cancer [1]. The most widely applied technique for detecting tumors is positron emission tomography (PET), a high-sensitivity imaging method with unlimited tissue penetration depth. The gold standard PET radiotracer used in the clinic is 2-deoxy-2-[^18^F]fluoro-D-glucose ([^18^F]FDG) [2]. [^18^F]FDG exploits the deregulation of cellular metabolism (the Warburg effect), which causes increased expression of GLUT1, the main facilitative hexose transporter for glucose [3,4]. However, there has been ongoing interest in developing more specific radiotracers, including radiotracers that target other transporter proteins for nutrients associated with metabolic processes or radiotracers targeting specific signatures expressed in tumor cells. Additionally, hypoxia is one of the leading reasons for treatment failure, so identifying hypoxic regions would be beneficial for patient care [5,6]. Some notable examples are 3′-deoxy-3′[^18^F]fluorothymidine ([^18^F]FLT) for monitoring nucleoside transporters/cell proliferation, [^18^F]FAZA and [^18^F]FMISO for identifying regions of hypoxia, ^68^Ga-NOTA-bombesin derivatives for imaging prostate cancer by targeting gastrin-releasing peptide receptors (GRPR), and many others [7,8,9,10].

Fluorescence imaging (FI) offers an alternative to PET, with particular applicability for in situ detection of cancer during surgical resection of solid tumors [11,12]. Fluorescence-guided surgery is advantageous because it allows accurate tumor margin determination in real time. Fluorescence-guided surgery would facilitate maximizing the amount of cancerous tissue removed while minimizing the loss of healthy tissue. This is especially important where the delineation between healthy and cancerous tissue is difficult to discern by simple visual inspection. Combining PET and FI techniques into a single dual probe would provide synergistic capabilities to improve patient outcomes. PET provides detection with high sensitivity, while FI provides opportunities for longitudinal imaging without the time constraints of a radioactive half-life. Dual probe would ensure there are no biodistribution differences between the two imaging modalities as the tracer is the same compound. The tumors would first be located using PET in the clinic, followed by fluorescence-guided surgical resection.

This short review will examine recent publications on dual PET/FI probes and comment on general molecular imaging probe design considerations. There are many scaffolds, platforms, and vectors on which dual probes can be built, so the many possibilities afford a wide variety of compounds. The several classes of compounds discussed can generally be organized under four groups: peptides, small molecules, antibodies, and nanoparticles.

## 2. Selection of PET Radionuclide

An essential initial decision when designing a PET imaging molecular probe is the choice of an appropriate radionuclide suitable for the intended purpose. Some of the most commonly used radionuclides discussed in this review include ^18^F, ^64^Cu, ^68^Ga, ^89^Zr, and ^124^I (Table 1) [13]. However, some recent advances have led to the development of more exotic radionuclides for research, such as ^44g^Sc, ^132^La, and ^133^La (Table 1) [14,15,16,17]. The physical half-life of the radionuclide should generally match the biological half-life of the radiotracer in vivo. For example, shorter half-life positron emitters (i.e., ^11^C, ^18^F, and ^68^Ga) are commonly used to label small molecules and peptides as these compounds are cleared by the body within minutes/hours [10]. While ^13^N and ^15^O find utility in blood-flow imaging, their short half-lives preclude their use for molecular imaging as dual probes [18,19,20]. As antibodies possess longer residence times in vivo, they are more suited to be labelled with longer-lived isotopes, such as ^64^Cu or ^89^Zr.

Another aspect of the radionuclide to consider is the energy of the emitted positron. The positron energy determines the spatial resolution of PET in vivo. The higher the positron energy, the longer the positron takes before an annihilation event, resulting in lower spatial resolution [21]. Another consideration is that the radioisotope’s branching ratio for positron emission should be high to maximize positron emission and limit exposure to other emission types. For example, the commonly used isotope ^64^Cu decays by positron emission, beta decay, electron capture, or internal conversion; this imparts unnecessary radiation dose that is not required for PET scans. Finally, the radioisotope should be of high radionuclidic purity.

The use of ^11^C and ^18^F generally requires covalent incorporation into a molecule, adding an additional layer of complexity in designing radiotracers as these radiosynthesis routes vary in length and difficulty. Additionally, the radiosynthesis time must be compatible with the physical half-life of the radionuclide.

With ^11^C having a short half-life of 20.4 min, its incorporation into a complex probe may prove to be difficult, as evidenced by the lack of dual probes using ^11^C in the literature. On the other hand, ^18^F has been employed quite regularly [22,23,24]. The radiochemistry of ^18^F has been widely explored to find suitable methods for rapid incorporation into radiotracers. The most common methods used are classic nucleophilic substitution S_N_2/S_N_Ar chemistry and ^18^F/^19^F isotopic exchange reactions [25,26,27]. Furthermore, ^18^F does not have to be covalently bound as it can be complexed to group III metals, such as Al and Ga, which are subsequently attached to the molecular probe by coordination chemistry [28,29,30]. However, this technique has not yet been applied for dual probes. 

Radiometals are typically incorporated into chelators to form kinetically and thermodynamically stable complexes. Prominent chelators include the macrocycles NOTA (1,4,7-triazacyclononane-1,4,7-triacetic acid) or DOTA (1,4,7,10-tetraazacyclododecane-1,4,7,10-tetraacetic acid) and their derivatives (Figure 1) [31]. These chelators are often incorporated through one of the carboxylic acids via amide linkages. There are also several other commercially available derivatives that allow for different incorporation methods (vide infra). The radiochemistry of radiometals is generally more straightforward than that of ^18^F. Incorporating the radiometal into the chelator at a late-stage step of the radiolabeling procedure under relatively mild conditions allows the use of sensitive compounds, such as peptides and/or antibodies. Common conditions include reacting NOTA/DOTA with the radiometal in a pH 4–5 buffer system. Moreover, large chelators, such as NOTA or DOTA, are also better suited for labelling larger molecules, such as peptides, as the influence of the chelator is comparatively less significant than that of a chelator-conjugated small molecule. Another factor to consider is that NOTA derivatives can commonly be labelled at room temperature, as opposed to DOTA derivatives, which require elevated temperatures up to 100 °C. This prompts the use of NOTA for more sensitive molecules, such as antibodies. Another advantage of radiometal coordination chemistry is that the purification step is often operationally simpler than that of ^18^F chemistry. Purifications can be optimized to only use solid-phase extraction cartridges instead of using high-performance liquid chromatography.

## 3. Fluorophores

The choice of fluorophore is crucial to ensure success of the design of dual-imaging probes. First, the excitation and emission wavelengths of the fluorophore should be biocompatible, meaning visible wavelengths (400–650 nm) or even near-infrared wavelengths (NIR) (>750 nm) [32,33]. Imaging NIR wavelengths allows greater tissue penetration while minimizing the autofluorescence of tissue and blood [34]. 

The most commonly employed NIR fluorophores are those of the cyanine dye family as many are commercially available. Recently, some organic dyes fluorescing in the NIR-II range (>1000 nm) have been described [35,36]. These NIR-II fluorophores have even greater tissue penetration depth while further minimizing background fluorescence, thus providing excellent signal-to-noise ratios for in vivo applications. The absorbance of many blood components and tissues drops off at 1000 nm [35]. However, these NIR-I and NIR-II fluorophores are usually rather large and could potentially interfere with the bioactivity of the probe, meaning the ability to bind to the cellular target or differential distribution (binding to plasma proteins, different cellular uptake, etc.). Thus, larger dyes are usually reserved for larger molecules, such as peptides, antibodies, and nanoparticles.

## 4. Peptides

Peptides are common scaffolds for designing dual-imaging probes because the conjugation of fluorescent prosthetic groups, chelators, and other radiolabeling motifs is less likely to negatively influence the biological activity compared to small molecules. This is especially true if a linker that positions these modifications away from the binding motif is utilized. While there are many advantages, one drawback with peptides is the relatively short half-life in vivo due to degradation by proteases [37]. Two common approaches to extend peptide half-life are incorporating unnatural amino acid residues into the chain or cyclizing the peptide.

Hübner et al. modified a homodimeric PESIN (PEG_3_-BBN_7–14_ [QWAVGHLM]) scaffold (for targeting gastrin-releasing peptide receptor, GRPR) with a fluorophore and incorporated a NODA-GA chelator for radiometal incorporation (Figure 2) [38]. The study looked into several different fluorophores to elucidate their influence on radiochemical yield, hydrophilicity, and biological activity of the PESIN compound. Among these fluorophores were coumarin, fluorescein, rhodamine, nitrobenzoxadiazole, dansyl, and pyridinium dyes, which were connected to the probe via the ε-NH_2_ group of a lysine residue. It was found that the more negative the overall charge of the molecule, the greater the decrease in binding affinity, resulting in higher IC_50_ values. This disfavored the use of fluorescein as a fluorophore. However, the type of fluorophore used had little impact on the radiolabeling efficiency. The most promising dual-probe compound contained a pyridinium dye (Figure 2), where **1a** absorbs at 480 nm and fluoresces at 635 nm. The positively charged pyridinium dye helped the dual probe retain biological function by reducing the overall negative charge. Confocal microscopy images were obtained using the pyridinium-containing dual probe, and cytosolic uptake was visualized in GRPR-positive PC-3 cancer cells. This indicated that compounds **1a** and **1b** were able to be internalized into the cells. While the pyridinium compound fluoresces well into the red region, the penetration depth could be improved by changing the fluorophore for one that emits at a longer wavelength.

Published in the same year, Hübner et al. also synthesized dual probes with the same homodimeric PESIN scaffold containing a NODA-GA but with three NIR cyanine fluorophores (Figure 2) [39]. These fluorophores were able to realize improved penetration depth in comparison to the other study [38]. The cyanine dyes varied in hydrophilicity and point of attachment. In analogy to the other study, the dyes that contained more sulfonic acid groups, and therefore possessed more negative charge, displayed a decrease in affinity for GRPR. Probe **1c** possessed an IC_50_ of 27.39 ± 2.01 nM for GRPR against [^125^I]I-Tyr^4^-bombesin, **1d** had an IC_50_ of 56.07 ± 1.47 nM, **1e** possessed an IC_50_ of 181.23 ± 2.24 nM, while the PESIN dimer that lacked the fluorophore possessed an IC_50_ of 21.48 ± 1.20 nM (compared to 2.81 ± 0.56 nM for BBN). These IC_50_ values for **1c** and the unmodified PESIN dimer are extremely similar, further exemplifying the possibility of using **1c** as a dual probe. Unfortunately, no in vivo studies were disclosed.

Zhang et al. also targeted prostate cancer using bombesin derivatives as GRPR antagonists [40]. By modifying DOTA-Lys-PEG_4_-[D-Phe^6^, Sta^13^]-BBN_6-14_NH_2_ with IRDye 650 *N*-hydroxysuccinimide (NHS) ester, they were able to generate **HZ220** as a metal-free dual-probe precursor (Figure 3). **^67/nat^Ga-HZ220** was validated in vitro with PC-3 cells and showed competitive inhibition against [^125^I]I-Tyr^4^-bombesin with an IC_50_ = 21.4 ± 7.4 nM. They were able to visualize internalization of **^67^Ga-HZ220** by confocal microscopy. **^68^Ga-HZ220** was also validated in vivo in PC-3 xenograft tumor-bearing mice, providing a tumor-to-muscle ratio of 25.8 ± 2.3 for fluorescence (calculated for ex vivo measurements) and 40 ± 11 for PET. The tumor-to-muscle ratio is an important metric in determining the suitability of a radiotracer. A high tumor-to-muscle ratio is indicative of good resolution and high specificity. These particular tumor-to-muscle ratios were on the higher end, showing that these modifications of the bombesin peptide were well tolerated.

Kasten et al. targeted MMP-14 (matrix metalloproteinase), which is overexpressed in glioma, with a novel dual probe that contained a NOTA for radiometal chelation and a NIR dye/quencher pair (Figure 4) [41]. Probe **2** possesses two different peptide sequences: “HWKHLHNTKTFL” is used as the MMP-14 binding motif, while “RS-Cit-G-HPhe-YLY” links the fluorescent quencher to the rest of the molecule. This latter sequence is cleaved by MMP-14, which causes the release of the dye quencher, allowing visualization of fluorescence. Therefore, probe **2** is more accurately described as a fluorogenic dual probe. This process allows improved signal-to-noise ratios as the fluorescence is recovered and amplified only in the presence of MMP-14. 

Dual probe **2** was radiolabeled with both ^68^Ga and ^64^Cu, and it was found that the **^64^Cu-2** substrate had more desirable properties. Due to the longer half-life of ^64^Cu (12.7 h) compared to ^68^Ga (67.7 min), **^64^Cu-2** was amenable to more prolonged PET analysis. Additionally, the conditions used to radiolabel with ^64^Cu were milder as they could be performed at room temperature for 20 min in pH 6.0 NaOAc buffer (87–91% yield). In comparison, the ^68^Ga labelling required heating at 90 °C for 20 min in pH 4.5 NaOAc buffer (100% yield). This dual probe showed promising preclinical results in PDX tumor-bearing mice for this NIR dye/quencher pair method, but there were still relatively low PET signals in comparison to other studies in this review. In athymic nude mice that had orthotopic PDX JX12 glioma tumors, **^64^Cu-2** had 0.43 ± 0.13% ID·g^−1^ (mean injected dose per gram, representing the ratio of tissue uptake vs. the injected dose). This lower value was hypothesized to be due to the lower expression of MMP-14 in comparison to other cancer biomarkers, among various other reasons. This method shows promise but needs to be further explored to confirm the utility of these large NIR dye/quencher pairs in addition to NOTA functionalization. 

Li et al. recently published a first-in-human study on the dual probe **^68^Ga-IRDye800CW-BBN** (Figure 5, top) [42]. As the name suggests, this modified bombesin peptide contains IRDye800CW as the fluorophore. It also possesses a NOTA as the ^68^Ga chelator. **^68^Ga-IRDye800CW-BBN** was first validated preclinically in mice using an orthotopic U87MG glioma xenograft model. Using intraoperative NIRF-guided resection, clear tumor margins in the orthotopic brain tumors were delineated and allowed the detection of tumor tissues that could not be visualized with the naked eye. 

The resected tissues were confirmed to be cancerous from pathological analyses (hematoxylin and eosin staining), which evidenced the effectiveness of this intraoperative NIRF method. During the clinical trial, the uptake of **^68^Ga-IRDye800CW-BBN** was compared to the previously established tracer ^68^Ga-NOTA-BBN_7–14_ as a PET imaging agent [43]. Scans were taken from the same patient, and it was shown that the two probes had almost identical biodistribution profiles; the dual probe had a tumor-to-background ratio of 13.43 ± 1.27 (based on the standardized uptake value, SUV_mean_), while the tracer ^68^Ga-NOTA-BBN_7–14_ had a tumor-to-background ratio of 12.12 ± 2.38. The patients then underwent NIRF-guided surgical resection of the glioblastoma tumors (Figure 5, bottom). Most importantly, the patients treated with this method achieved 80% PFS-6 (six-month progression-free survival), which was a marked improvement compared to 46% for patients who underwent 5-ALA (5-aminolevulinic acid)/fluorescein-guided surgical resection [44]. The success of this particular dual probe is likely due to the modification of the peptide terminus while retaining the proposed binding motif required for biological activity. There is a lot of interest in this dual probe and high hopes that these PFS-6 values remain high as the sample size for clinical trials increases. Hopefully, these positive results will set the stage for more dual probes to be approved and implemented.

Fan et al. designed a dual probe based on the porphyrin derivative pyropheophorbide-a (Pyro) (Figure 6) [45]. Pyro possesses both chelation properties and fluorescent properties, allowing chelation of a radiometal and optical imaging in one motif. This is advantageous in that the peptide only needs to be functionalized once by incorporating the bifunctional Pyro motif. Pyro was conjugated with an RGD dimer to achieve tumor-specific accumulation in glioblastoma, forming **P3PRGD_2_** (Pyro-3PEG_4_-E[c(RGDfK)]_2_), which was subsequently labelled with ^64^Cu. Fan et al. previously developed the novel RGD dimer c(RGDfK)_2_ and found that it specifically bound integrin α_v_β_3_ better than the monomer or other multimers. Several PEG linkers were incorporated into the dual probe to improve hydrophilicity, rendering the probe water-soluble. **P3PRGD_2_** was first validated in vitro by confocal microscopy using U87MG cells. Fluorescence was visualized (has 702 nm emission but used a 740 nm filter) in U87MG cells and showed a blocking effect by the nonfluorescent RGD dimer c(RGDfK)_2_, showing that **P3PRGD_2_** was still selective for integrin α_v_β_3_. **P3PRGD_2_** was also subjected to in vivo optical imaging, which reflected the results obtained in the in vitro experiments. Next, **P3PRGD_2_** was radiolabeled with ^64^Cu using a ^64^CuCl_2_ solution in 0.1 M NH_4_OAc buffer at 100 °C for 30 min. **^64^Cu-P3PRGD_2_** was imaged in mice implanted with subcutaneous U87MG tumors, providing 5.61 ± 0.86% ID·g^−1^ one hour post injection, but with higher levels in the liver and kidneys. The uptake of the probe showed a blocking effect when coinjected with excess cold peptide, further confirming the specificity of this dual probe.

Sun et al. developed **^68^Ga-SCH2** as a dual NIR-II fluorescent and PET probe for integrin α_v_β_3_ imaging (Figure 7) [36]. The probe has an absorption band between 600 and 900 nm and a maximum emission wavelength of 1055 nm. An advantage of the fluorophore they used was that it exhibited high photostability in buffer and in mouse serum compared to the clinically approved ICG (indocyanine green) dye. The radiolabeled **^68^Ga-SCH2** also exhibited high kinetic stability in mouse serum as there was no release of free ^68^Ga after 2 h. The dual probe showed favorable uptake in U87MG cells in vitro and in U87MG-tumor-bearing mice. The NIR-II fluorescent imaging showed a tumor-to-background ratio of 4.75 ± 0.22 at 12 h post injection, and the PET imaging showed 2.48 ± 0.32% ID·g^−1^ one hour post injection. In comparison to the work by Fan et al. involving **^64^Cu-P3PRGD_2_** [45], **^68^Ga-SCH2** has a lower ID·g^−1^ value. The core peptide dimers are slightly different, but the main reason for this difference in uptake could be due to the size of the dual probe modifications on **^68^Ga-SCH2**.

Sun et al. also introduced the concept of using photo-click chemistry to incorporate a pyrazoline-based fluorophore and a NOTA chelator in one step (Figure 8) [46]. The dual probe possessed a maximum absorption at 360 nm and maximum emission at 570 nm, giving a large Stokes shift. A tetrazole containing the NOTA was activated under 254 nm light and then reacted with an alkene-functionalized peptide AE105 to produce dual probe **3**. The AE105 peptide targeted uPAR (urokinase-type plasminogen activator receptor), a serine protease that is a biomarker in cancer development [47]. The U87MG cell line was used in their studies as they overexpress uPAR. Fluorescence was visualized in vitro, and a blocking effect was observed with excess AE105. After labelling **3** with ^68^Ga, U87MG tumor-bearing-mice were imaged using PET. A high tumor-to-muscle ratio was observed with ~5% ID·g^−^^1^. This work provided a great proof-of-concept for the use of photo-click chemistry to rapidly incorporate a dual-probe moiety. Nevertheless, the fluorophore was unfortunately not as biologically relevant as NIR fluorophores, such as their work with **^68^Ga-SCH2**. This issue could potentially be rectified by the use of highly conjugated pyrazolines to increase the emission wavelength [48].

## 5. Small Molecules

In order to enable dual-probe capability in small molecules, careful design must be employed to ensure that the molecule retains selectivity for the biomarker while containing both a fluorophore and a radionuclide. Because of these restrictions, large fluorophores and/or chelators are often not incorporated due to their greater influence on the bioactivity of small molecules. However, there are always some compounds that violate these general guidelines.

An et al. developed a dual probe based on a Cy5 fluorophore and an aryltrifluoroborate prosthetic group (Figure 9, left) [49]. The probe was able to stain A549 cells (human epithelial lung carcinoma) in vitro in a concentration-dependent manner. **[^18^F]Cy5-BF_3_** was selectively taken up in A549 xenograft tumors in mice, which was confirmed by PET and by ex vivo biodistribution. Zhang et al. also developed a dual probe that was based on a cyanine dye (MHI-148) that contained a DOTA chelator motif (Figure 9, right) [50]. In an in vivo hepatocellular carcinoma (HCC) model, **DOTA-MHI-148** showed selective uptake in the tumor under NIRF and PET imaging (once radiolabeled with ^68^Ga). The high mitochondrial membrane potential of tumor cells relative to normal cells leads to higher accumulation of polymethine dyes [51]. Additionally, hypoxia and OATPs (organic anion transporter polypeptides) have been postulated to be responsible for the uptake of polymethine dyes by cancer cells despite their apparent lack of targeting motifs [50,51]. It would be of interest to further explore these seemingly specific yet not specific dyes and investigate their mechanism of tumor localization.

Baranski et al. synthesized several different prostate-specific membrane antigen (PSMA) inhibitors (Figure 10, top) [52]. PSMA is overexpressed in several types of prostate cancer [53]. The dual probe builds off of the commonly used PSMA-11 motif, which consists of glutamic acid and lysine linked by a urea motif. A new PET tracer that targets PSMA was recently approved, so there is good precedent for its use as a biomarker [54]. The chelator used to bind ^68^Ga was *N*,*N′*-bis[2-hydroxy-5-(carboxyethyl)benzyl]ethylenediamine-*N*,*N′*-diacetic acid (HBED-CC), which was accomplished in >99% radiochemical yields. The authors tested four different fluorophores: FITC, AlexaFluor488, IRDye800CW, and DyLight800. The in vitro and in vivo experiments showed that all four dual probes showed uptake into LNCaP cells/xenograft tumors, while there was no uptake in PSMA-free PC3 cells. Moving forward, dual probes that possessed NIR fluorophores (IRDye800CW and DyLight800) were successfully used in a proof-of-concept fluorescence-guided surgical resection.

Aras et al. also created a dual probe based on the PSMA-11 motif (Figure 10, bottom) [55]. **[^18^F]-BF_3_-Cy3-ACUPA** was validated in an in-human study with PSMA-positive tumors. This probe used a Cy3 dye as the fluorophore (maximum excitation: 554 nm, maximum emission: 565 nm) and an ammonium methyl trifluoroborate (AMBF_3_) as the ^18^F-containing component. The ^18^F was incorporated by an isotopic exchange reaction, and the resulting molar activity was greater than 0.116 ± 0.077 mCi/nmol, which is quite exceptional for an isotopic exchange. Moreover, this method of ^18^F-incorporation is operationally simpler than other classical methods. There was minimal skeletal uptake, thus indicating that AMBF_3_ was resistant to in vivo defluorination. In addition to providing high primary tumor-to-blood ratios (TBR 2.8–32.5), **[^18^F]-BF_3_-Cy3-ACUPA** had an exceptional ability to identify small metastases with TBR between 3.6 and 50.2. As there were no commercially available devices for Cy3-specific imaging, Aras et al. had to develop a custom imaging device to enable intraoperative fluorescence imaging. From a small cohort of 10 patients, the results were extremely positive, and it will be exciting to see this work expand into a larger clinical trial.

Ortmeyer et al. developed compound **[^18^F]4**, a ^18^F-labelled caspase inhibitor containing a BODIPY fluorophore (Figure 11) [22]. Due to the lower emission wavelength compared to cyanine dyes (λ = 509 nm vs. >700 nm), **[^18^F]4** was not suitable for in vivo fluorescence imaging. However, this dual probe showed nanomolar-level IC_50_ values against caspase-3 and caspase-7 in an in vitro assay. The authors are currently exploring the use of a longer spacer between the bioactive isatin moiety and the fluorophore and determining whether this improves the bioactivity. The biodistribution of **[^18^F]4** was tested in vivo in adult C57/Bl6 mice. It was found that the tracer accumulated in high amounts in the liver, while there was less radioactivity accumulated in the kidneys/bladder. The metabolic stability in serum was subsequently evaluated, and it was found that the tracer remained mostly stable over the course of 2 h with around ~5% degradation. This was consistent with the small amount of bone uptake that was seen. Despite the described in vivo work, no data regarding animal tumor models was reported. 

Allott et al. derivatized a previously reported glycogen-binding fluorescent probe (CDg4) to contain a ^18^F, providing dual probe **[^18^F]5** (Figure 11) [23]. Glycogen reporters have the advantage of being able to detect metabolically depressed quiescent cancer cells as the glycogen stores promote survival during times of nutrient stress [56]. **[^18^F]5** absorbs at 420 nm and fluoresces at 550 nm, optical properties that disfavor its use in patients. Radiosynthesis was achieved in a three-step synthesis to provide **[^18^F]5** in 5.1 ± 0.9% n.d.c RCY (nondecay corrected radiochemical yield) with molar activity of A_m_ = 7.6 GBq/µmol. By fluorescence microscopy, this dual probe showed uptake in glycogen-containing cancer cell lines and showed no uptake in cells that were treated with α-amylase (to degrade glycogen). However, there was no correlation between glycogen levels and uptake of **[^18^F]5**. Additionally, the compound showed metabolic instability in vivo, likely due to the presence of the Michael acceptor moieties. Unfortunately, this particular method where the dual-probe moiety was also the pharmacophore is not widely applicable.

Our group recently published the synthesis and evaluation of several different dual probes targeting GLUT1, with **2-[^18^F]FBDG** being the most promising (Figure 11) [24]. The probe was inspired by the commonly used fluorescent probe 2-NBDG (2-(*N*-(7-nitrobenz-2-oxa-1,3-diazol-4-yl)amino)-2-deoxyglucose). By replacing the nitro group with a fluorine-containing group, 2-NBDG can become a dual probe. The sulfonyl fluoride was operationally simple to install nonradiochemically and radiochemically, with a 69 ± 3% d.c. RCY (decay-corrected radiochemical yield) over two steps. **2-[^18^F]FBDG** possesses an absorption wavelength of 425 nm and an emission wavelength of 570 nm. While **2-FBDG** showed good uptake in EMT6 and MDA-MB231 cells by in vitro fluorescence microscopy experiments, **2-[^18^F]FBDG** unfortunately did not show significant uptake in vivo into MDA-MB231 xenograft tumors by PET imaging. Some activity was trapped in the tumor, but there was a significant amount of defluorination in vivo as evidenced by the bone uptake. The sulfonyl fluoride was regrettably not stable in vivo, but there is definitely room for improving these GLUT1-targeting tracers. While this sulfonyl fluoride fluorophore was not suitable for PET imaging, it could still potentially be used as a moiety for fluorescent imaging.

## 6. Antibodies

Antibodies can display high specificity for their cellular targets and have long residence times in vivo. These properties render antibodies highly attractive candidates for radiolabeling with longer-lived radionuclides, such as ^64^Cu and ^124^I. However, one of the main issues regarding antibodies is that they are quite difficult to fully characterize. No two batches of derivatized antibody will be exactly the same. Special care must be taken to ensure that the relative ratios of fluorophore/radiolabel are similar between batches, but this is difficult to completely verify as mass spectrometry is one of the few tools available for characterization. Several different studies have been performed with dual probe functionalized antibodies/antibody relatives.

Zhang et al. developed a **XB1034-cetuximab-TCO** conjugate, where XB1034 is a thiopyrylium NIR-II fluorophore and TCO is a reactive trans-cyclooctene (Figure 12) [57]. Cetuximab is a monoclonal antibody (~150 kDa) that confers selectivity for epidermal growth factor receptor (EGFR), which is often overexpressed in several types of cancers [58,59,60]. With MDA-MB231 or MCF7 xenograft tumor-bearing mice, they utilized a pre-targeting approach where the **XB1034-cetuximab-TCO** was injected, and the probe was allowed to incubate in vivo for 48 h. Then, **^68^Ga-NETA-Tz** was injected, and the tetrazine moiety underwent an inverse electron demand Diels–Alder reaction with the TCO moiety of the mAb. Unreacted **^68^Ga-NETA-Tz** showed rapid clearance via the kidneys with a decrease from 1.49 ± 0.21% to 0.77 ± 0.05% ID·g^−1^ in the blood. This minimized the high background and provided ~1% ID·g^−1^ of the final dual probe. This value seems low, but the tumor-to-background ratio at 48 h for the NIR-II imaging was still adequate at ~7, showing that the pretargeting approach improved the overall S/N ratio. After sufficient decay of the ^68^Ga isotope (10 half-lives), NIR-II fluorescence-guided surgery occurred. The probe had a maximum absorption at 982 nm and a maximum emission at 1044 nm. The authors actually used 808 nm excitation with an improvised 1000 nm long-pass filter for their in vivo NIR-II imaging. Uptake of the tracer into MDA-MB231 tumors showed sufficient blocking by cetuximab coinjection in both NIR-II and PET imaging, signifying the specificity of this method. MCF7 tumors had quite low uptake, and the level of radioactivity was comparable to that of the MDA-MB231 blocking experiments. 

Wang et al. used a P-glycoprotein (Pgp) antibody to image chemoresistant tumors [61]. Pgp is an ATP-binding cassette (ABC) transporter that facilitates the efflux of cytotoxic drugs from cancer cells. The anti-Pgp mAb (Pab) was reacted with excess DOTA-sulfo-NHS followed by excess IR800-NHS to introduce a DOTA chelator and a fluorescent cyanine dye, forming **DOTA-Pab-IR800** (Figure 13). Using NCI/ADR-Res (chemoresistant ovarian cancer cell line) xenograft tumors in mice, **^64^Cu-DOTA-Pab-IR800** was validated to have 9.9 ± 1.4%, 12.1 ± 1.2%, and 10.5 ± 1.0% ID·g^−1^ at 4, 24, and 48 h post injection. This shows that the signal is stable over 48 h and is a great method for longitudinal imaging. The control **^64^Cu-DOTA-IgG** tracer showed 6.2 ± 0.8%, 7.2 ± 1.1%, and 6.0 ± 2.0% ID·g^−1^ at 4, 24, and 48 h. There was also high liver and kidney uptake of the tracer, but eventual washout was observed. Interestingly, by fluorescent imaging, no liver/kidney uptake was observed and only the tumor was visible. The authors mentioned that these internal organs showed little or no fluorescence signal due to the attenuation of the light from scattering/absorption. The xenograft tumors were subcutaneous, so they were able to be visualized.

Zettlitz et al. developed an antiprostate stem cell antigen (PSCA) A2 Cys-diabody (A2cDb) that was conjugated with IRDye800CW and labeled nonspecifically with ^124^I (Figure 14) [62]. PSCA is a cell surface glycoprotein that is overexpressed in most prostate cancers but is also overexpressed in most pancreatic ductal adenocarcinomas [63]. The disulfide bridge of the A2cDb was reduced using tris(2-carboxyethyl)phosphine (TCEP), and the free thiols were conjugated to IRDye800CW maleimide, causing site-specific labeling with the fluorophore. This technique ensured that the fluorophore would not interfere with the binding motif as the thiols were on the opposite side of the diabody. Then, the whole complex was subject to Pierce iodination tubes (which contains oxidizing reagents) with [^124^I]NaI to nonselectively iodinate tyrosine residues, forming **^124^I-A2cDb-800**. This was subjected to in vivo immune-PET experiments in patient-derived pancreatic ductal adenocarcinoma xenograft tumor-bearing mice followed by NIRF imaging. The **^124^I-A2cDb-800** dual probe showed selective targeting of the tumor with 0.28 ± 0.05% ID·g^−1^ in low-PSCA-level tumors and 1.02 ± 0.47% ID·g^−1^ in high-PSCA-level tumors. 

Wang et al. developed an Affibody containing a DOTA chelator and Cy5.5 as the fluorescent dye (Figure 15) [64]. 

Affibodies are nonimmunoglobulin proteins possessing a three-helix bundle. As antibody mimics, Affibodies possess favorable in vivo characteristics as they retain high specificity while having rapid blood clearance, contrary to antibodies. 

Wang et al. connected the Affibody to a poly(amido amine) (PAMAM) dendrimer via a cysteine–maleimide linkage. From there, the dendrimer was derivatized with Cy5.5-NHS and DOTA-NHS, then labeled with ^64^Cu to create **^64^Cu-DPCZ**. Using SKOV3 xenograft tumor-containing nude mice, NIRF and PET imaging was performed to validate the tracer. While NIRF imaging showed that the tracer continued to accumulate in the tumor over time, PET imaging showed that the tracer signals decreased after longer time points. A possible explanation for this divergence is the instability of the ^64^Cu-DOTA complex upon in vivo reduction of Cu(II) to Cu(I). Both imaging modalities showed both liver and kidney uptake.

## 7. Nanoparticles

Many nanoparticles described in the literature rely on the enhanced permeability and retention effect (EPR) to enter cancer cells [65], and many of these “nontargeting” nanoparticles that lack some sort of targeting vector do make it past clinical trials [66]. In contrast, nanoparticles that possess targeting vectors have largely not been able to succeed for reasons beyond the scope of this review [66]. There has not been much activity in developing a dual-probe nanoparticle, but one “nontargeting” and one “targeting” approach will be discussed in this section. 

Zhang et al. developed a nanoparticle capable of both NIR-II and PET imaging [67]. This group used their previously developed Mdot, which is a melanin-derived nanoparticle, to chelate ^64^Cu. This Mdot is associated with a mesoporous silica nanoparticle via electrostatic interactions, which is then protected with a lipid bilayer. These nanoparticles were then sonicated with **CH-4T** (Figure 16A), which is a benzobisthiadiazole dye, to incorporate the fluorophore into the lipid bilayer (Figure 16B). The fluorescence intensity of the fluorophore incorporated in the nanoparticle was about 4 times greater than that of **CH-4T** in solution alone. This phenomenon was attributed to the fluorescent dye interacting strongly with the supported lipid bilayer. This dual probe was evaluated in vivo using A431 (human epidermoid carcinoma) xenograft tumors in mice and showed the highest uptake at 8 h post injection (tumor-to-background = 4.43 ± 0.18 for NIR-II imaging with 1000 nm long-pass filter) along with high liver and spleen uptake (Figure 16C). PET imaging also showed the highest uptake at 8 h post injection (~14% ID·g^−1^) with the highest contrast at ~24 h (~8% ID·g^−1^) post injection as the nanoparticle displayed washout in the heart, liver, and spleen (Figure 16D). From the images, it is evident that there is more than sufficient contrast to be able to differentiate the tumor from the background.

Tang et al. developed silica nanoparticles that were conjugated with aptamers that were subsequently functionalized with cyanine dyes and DOTA chelators (Figure 17) followed by radiolabeling with ^64^Cu [68]. Aptamers are single-stranded oligonucleotides that can bind to their appropriate targets. However, aptamers could potentially be cleaved in vivo by nucleases and lose their intended bioactivity [69]. The size of the silica nanoparticles was readily controlled, and these authors tested both 20 and 200 nm nanoparticle sizes. The nanoparticles were functionalized with a 26-mer G-rich DNA aptamer that possessed a high binding affinity for nucleolin (overexpressed in the cytoplasm and on the outer surface of the plasma membrane of some cancer cells) [70]. The 20 nm aptamer functionalized nanoparticle showed uptake in vivo with 4T1 (murine breast cancer) xenograft tumors in mice. Notably, this allowed detection of lymph node metastases. The 200 nm nanoparticle remained highly localized at the injection site and was not taken up by the tumors, likely due to their large size.

## 8. Conclusions

Following this exploration of the different dual PET/FI probes in the literature, the future of dual probes appears to be moving towards the use of macrocyclic chelators as a facile way of incorporating radiometals and the use of NIR-I/NIR-II fluorophores to maximize penetration depth. Peptides are the most common scaffold upon which this dual-modality framework can be built. Peptides are advantageous in that many biomarkers can be readily targeted with peptides. They are relatively straightforward to characterize, and the chemical and biological properties can be readily modified. Despite the common occurrence of peptides, the other categories of dual probes show much promise as well. The main advantage of using small molecules comes with their ease of characterizability and chemistry, antibodies with their outstanding specificity, and nanoparticles with their seemingly unlimited modification. The increasing commercial availability of chelators and fluorophores will hopefully garner more interest in the field of dual probes. With the advent of the first-in-human clinical trial of **^68^Ga-IRDye800CW-BBN**, the concept of dual probes will become more widespread and will be further explored as a superb precision medicine concept. Hopefully, the trend in the coming years is an increase in clinical approval and translation of a large variety of dual probes as there are many more biomarkers that can be targeted. The application of precision medicine will definitely benefit from the availability of many different dual probes where the most optimal biomarker is targeted for each patient.

## Figures and Tables

**Figure 1 pharmaceutics-14-00645-f001:**
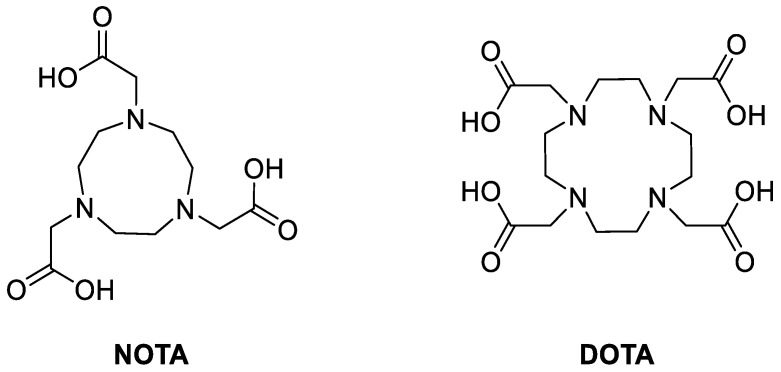
Structures of NOTA and DOTA.

**Figure 2 pharmaceutics-14-00645-f002:**
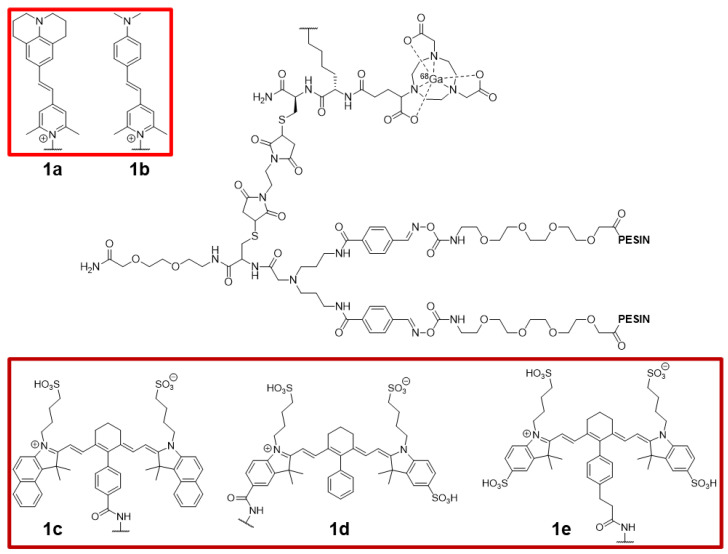
Structures of homodimeric PESIN-based dual probes **1a**–**e** for GRPR imaging.

**Figure 3 pharmaceutics-14-00645-f003:**
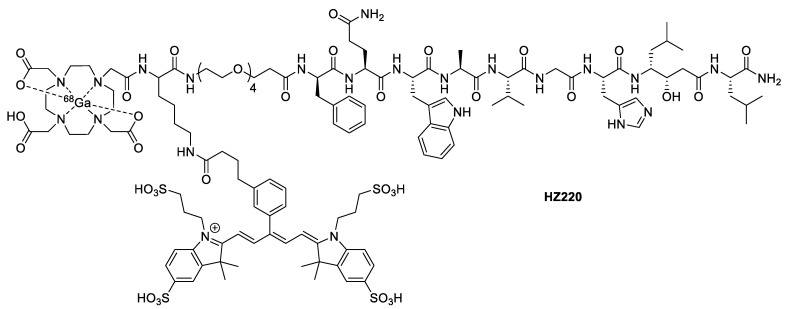
Structure of **^68^Ga-HZ220**.

**Figure 4 pharmaceutics-14-00645-f004:**
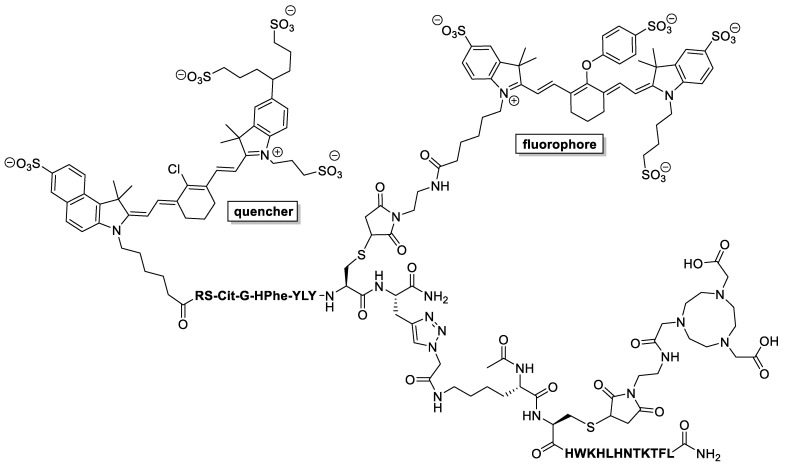
Structure of metal-free dual-probe precursor **2**, which targets MMP-14 as a glioma tracer.

**Figure 5 pharmaceutics-14-00645-f005:**
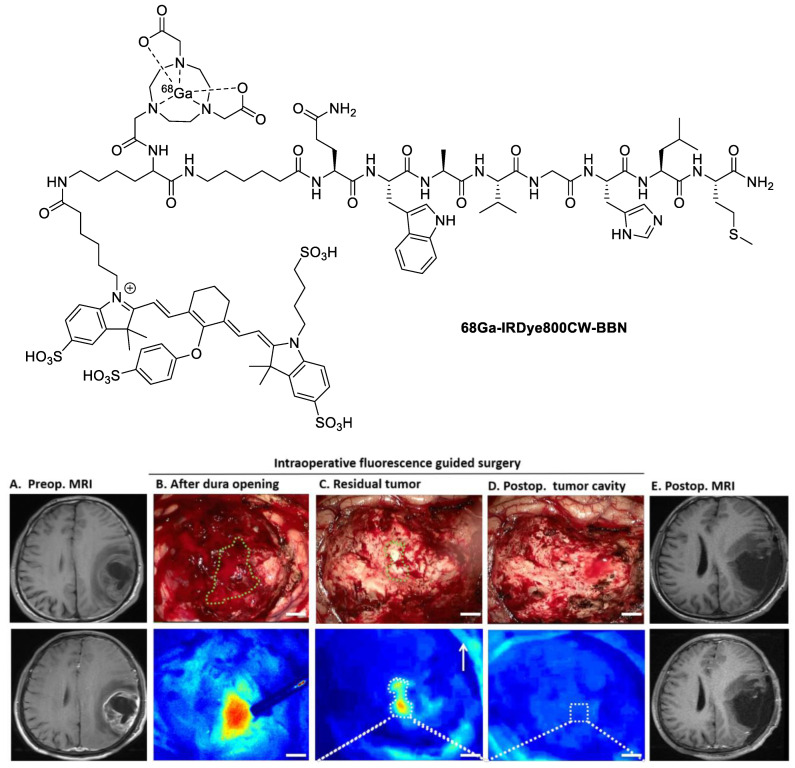
Structure of **^68^Ga-IRDye800CW-BBN** (**top**). NIRF image-guided glioblastoma resection using **^68^Ga-IRDye800CW-BBN** (**bottom:** Reprinted from Ref. [42]. 2018, Theranostics). The fluorescent tumor is delineated by green dashed lines in the white-light images.

**Figure 6 pharmaceutics-14-00645-f006:**
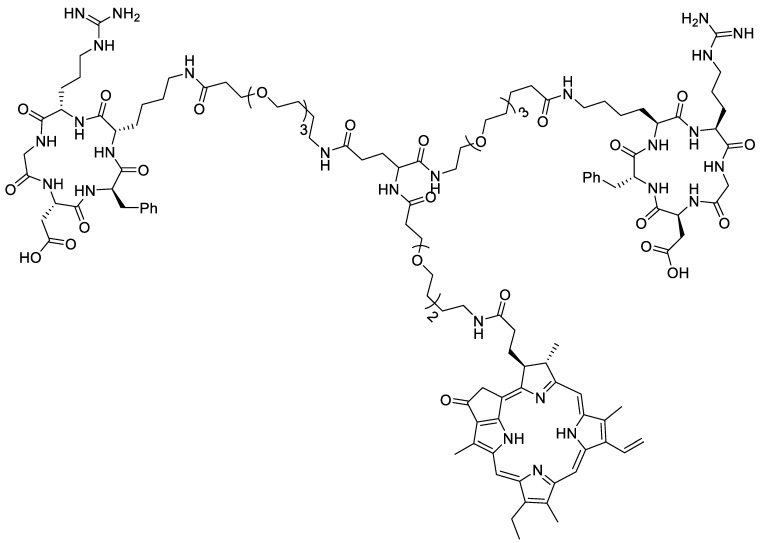
Structure of **P3PRGD_2_** = Pyro-3PEG_4_-E[c(RGDfK)]_2_.

**Figure 7 pharmaceutics-14-00645-f007:**
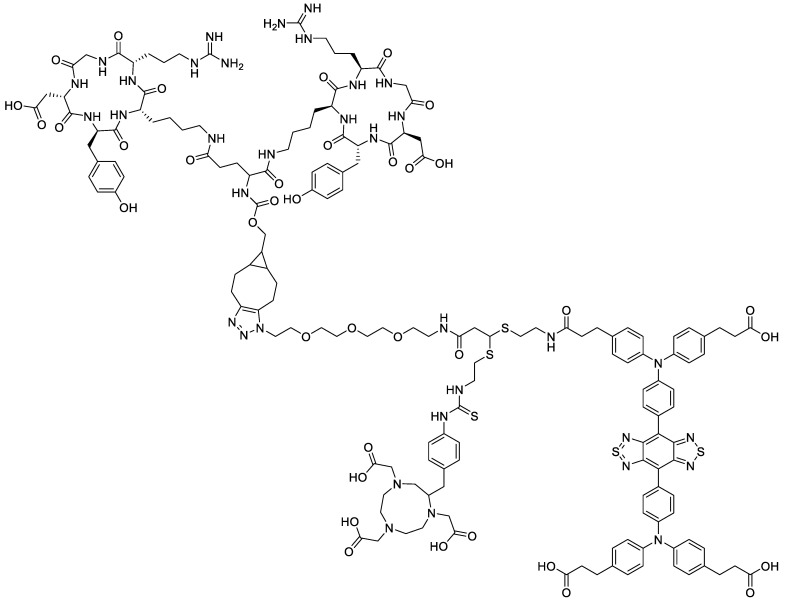
Structure of **SCH2**.

**Figure 8 pharmaceutics-14-00645-f008:**
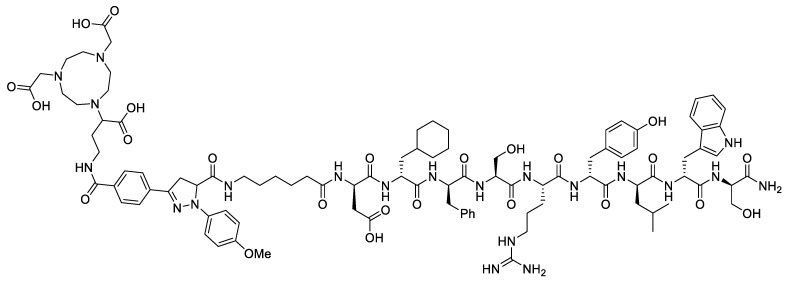
Structure of dual probe **3**.

**Figure 9 pharmaceutics-14-00645-f009:**
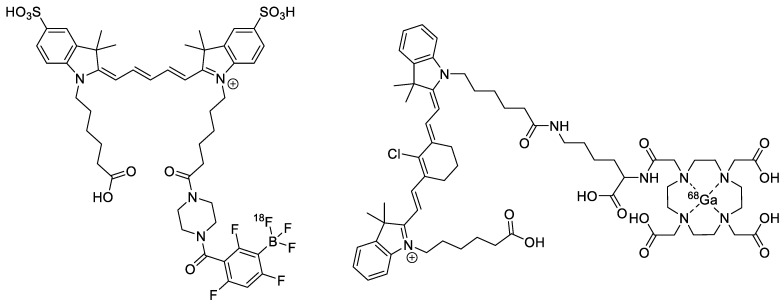
(**Left**) **[^18^F]Cy5-BF_3_**; (**right**) **[^68^Ga]DOTA-MHI-148**.

**Figure 10 pharmaceutics-14-00645-f010:**
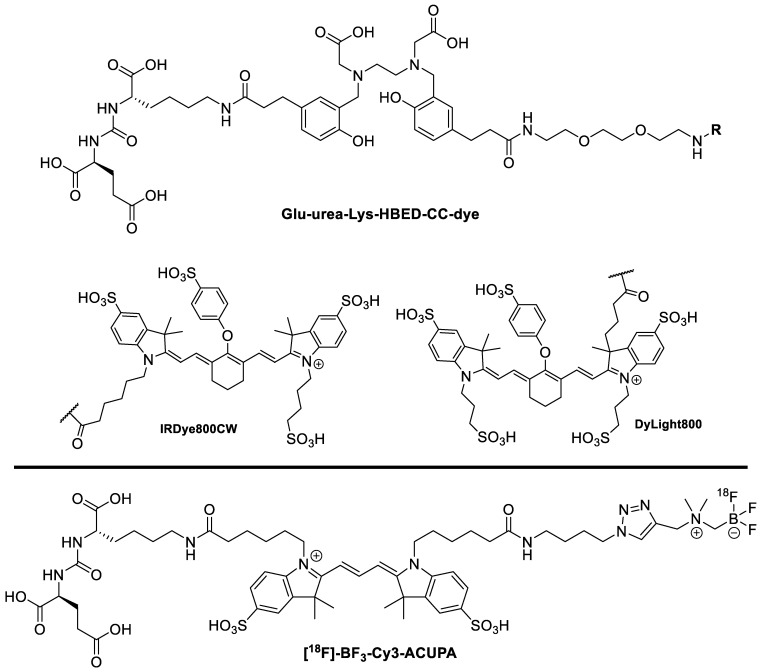
PSMA inhibitor compounds. (**Top**) Glu-urea-Lys-HBED-CC-IRDye800CW: R = IRDye800CW. **Glu-urea-Lys-HBED-CC-DyLight800**: R = DyLight800. (**Bottom**) **[^18^F]-BF_3_-Cy3-ACUPA**.

**Figure 11 pharmaceutics-14-00645-f011:**
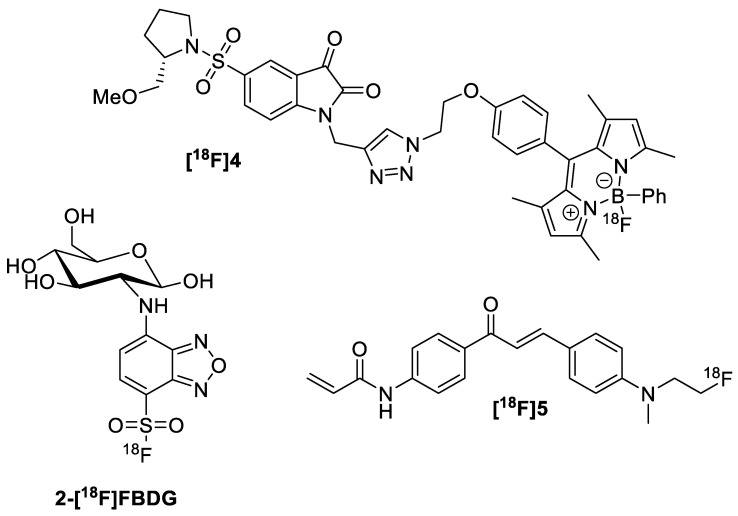
Non-cyanine-based small molecule dual probes: **[^18^F]4**, **[^18^F]5**, and **2-[^18^F]FBDG**.

**Figure 12 pharmaceutics-14-00645-f012:**
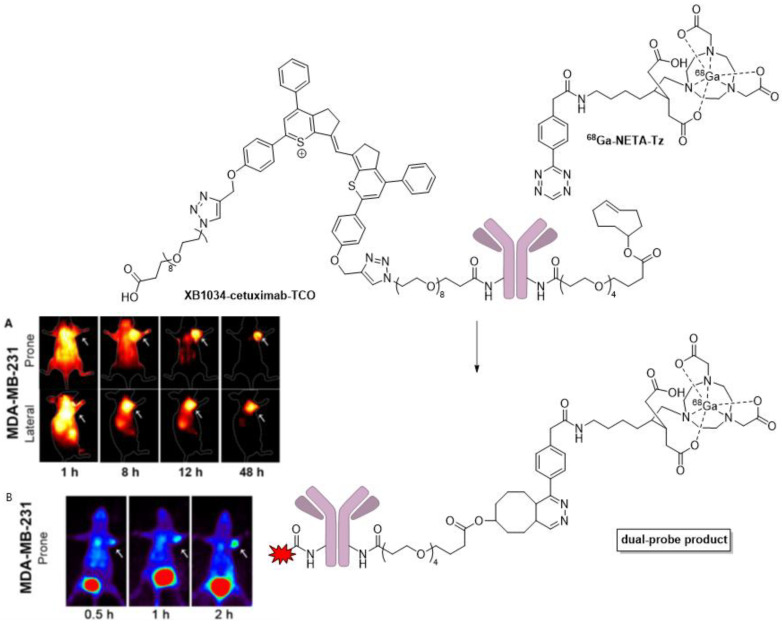
Structures of **XB1034-cetuximab-TCO** and **^68^Ga-NETA-Tz**. NIRF imaging (**A**) and PET images (**B**) in MDA-MB231 xenograft tumors at various time points after **^68^Ga-NETA-Tz** injection (Reprinted from ref. [57]. Copyright 2020 FEBS Press and John Wiley & Sons Ltd.)

**Figure 13 pharmaceutics-14-00645-f013:**
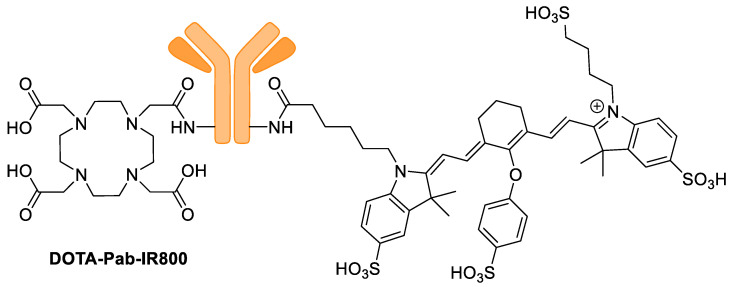
Structure of **DOTA-Pab-IR800**.

**Figure 14 pharmaceutics-14-00645-f014:**
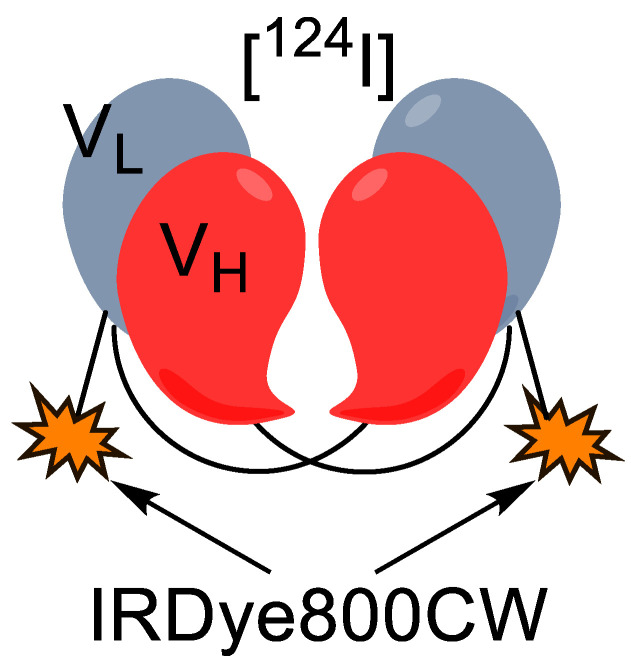
Diagram of **^124^I-A2cDb-800**.

**Figure 15 pharmaceutics-14-00645-f015:**
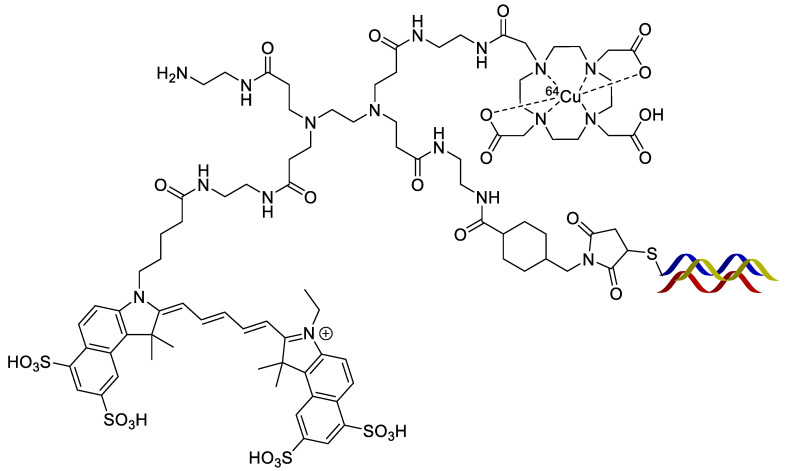
Structure of the Affibody **^64^Cu-DPCZ**.

**Figure 16 pharmaceutics-14-00645-f016:**
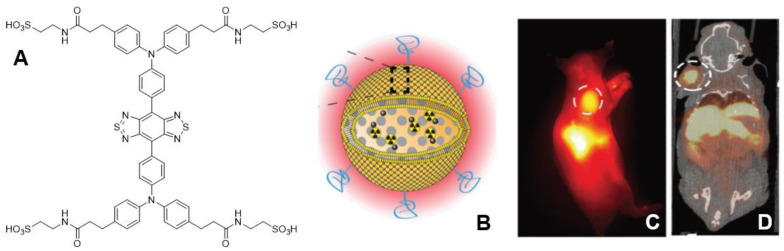
(**A**) Structure of the NIR-II dye **CH-4T**. (**B**) Schematic of overall dual-probe nanoparticle. (**C**) NIRF image in A431 xenograft tumor-bearing mouse. (**D**) PET/CT image (Reprinted from ref. [67]. Copyright 2019 WILEY-VCH Verlag GmbH & Co. KGaA, Weinheim.).

**Figure 17 pharmaceutics-14-00645-f017:**
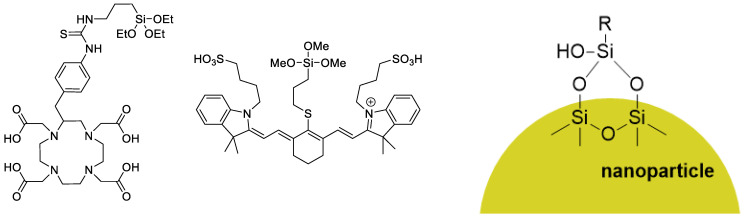
Structures of the silylated DOTA and cyanine derivatives used to functionalize the silica nanoparticles (**left**). Example of how the silyl groups interact with the silica nanoparticle surface (**right**).

**Table 1 pharmaceutics-14-00645-t001:** Typical positron emitters, half-lives, and mean positron energies [13].

Radionuclide	Half-Life	Mean Positron Energy (MeV)	Positron Emission Branching Ratio
^11^C	20.4 min	0.386	0.998
^13^N	9.97 min	1.199	0.998
^15^O	122.24 s	1.732	0.999
^18^F	109.7 min	0.250	0.967
^44g^Sc	4.0 h	0.632	0.943
^64^Cu	12.7 h	0.278	0.174
^68^Ga	67.7 min	0.829	0.891
^89^Zr	78.4 h	0.396	0.220
^124^I	100.2 h	0.214	0.260
^132^La	4.8 h	1.29	0.412
^133^La	3.9 h	0.461	0.072

## Data Availability

Not applicable.

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
