# Peer review of "Dual Probes for Positron Emission Tomography (PET) and Fluorescence Imaging (FI) of Cancer"

_pharmaceutics, 2022, doi:10.3390/pharmaceutics14030645_

Round 1

Reviewer 1 Report

General comments:

The subject of the paper is topical and of interest to the imaging community and includes a large amount of information relevant to the topic.  

However, my major issue with this review is that it reads like an inventory of various studies, without a critical appraisal or discussion of their results. The data is presented just as facts / values and the meaning/interpretation of the data in the context of tumour imaging using dual probes is left in the air.

This aspect must be corrected throughout the paper.

Other comments:

  1. Introduction: Next to FLT which is the most common PET tracer used for the evaluation of tumour proliferation, the authors should mention PET tracers for tumour hypoxia (FAZA, FMISO). Hypoxia and proliferation are two of the main culprits for treatment failure / tumour recurrence, thus PET can play a critical role in identifying those patients that would benefit the most from hypoxia-targeting or proliferation-targeting therapies. I suggest some recent reviews covering the above aspects:

https://pubmed.ncbi.nlm.nih.gov/34557414/

https://pubmed.ncbi.nlm.nih.gov/29907486/

  1. Section 2 – selection of PET radionuclides. Beside the listed physical and chemical properties required for PET isotopes, the authors should also mention a few additional requirements:
  • The isotope should preferably decay by pure positron emission so there is no additional exposure to the patients from potential electron emission
  • High radionuclidic purity.
  1. Table 1 should also include O-15 (used for PET evaluation of oxygen metabolism and blood flow) and N-13 (also for blood flow imaging).

Specific comments:

  1. Line 112 – “However, these fluorophores are …” – please check this phrase and reword for clarity
  2. Line 120 – “…is less likely to negatively influence the biological…small molecules.”
  3. Line 191 – replace ‘could not be visualized with the eyes alone’ with ‘could not be visualized with the naked eye’

Author Response

We thank the reviewer for the guidance and their valuable comments to improve the quality of our submission. Please find our responses below:

General comments:

The subject of the paper is topical and of interest to the imaging community and includes a large amount of information relevant to the topic.  

However, my major issue with this review is that it reads like an inventory of various studies, without a critical appraisal or discussion of their results. The data is presented just as facts / values and the meaning/interpretation of the data in the context of tumour imaging using dual probes is left in the air.

This aspect must be corrected throughout the paper.

Thank you for this analysis. We have corrected this in this revised manuscript. More comments have been made. There were already some introductory comments in the opening paragraphs of each section, we can see how more specific comments could be beneficial.

Other comments:

  1. Introduction: Next to FLT which is the most common PET tracer used for the evaluation of tumour proliferation, the authors should mention PET tracers for tumour hypoxia (FAZA, FMISO). Hypoxia and proliferation are two of the main culprits for treatment failure / tumour recurrence, thus PET can play a critical role in identifying those patients that would benefit the most from hypoxia-targeting or proliferation-targeting therapies. I suggest some recent reviews covering the above aspects:

https://pubmed.ncbi.nlm.nih.gov/34557414/

https://pubmed.ncbi.nlm.nih.gov/29907486/

Thank you, and we agree with this sentiment. Hypoxia has now been mentioned in the introduction.

  1. Section 2 – selection of PET radionuclides. Beside the listed physical and chemical properties required for PET isotopes, the authors should also mention a few additional requirements:
  • The isotope should preferably decay by pure positron emission so there is no additional exposure to the patients from potential electron emission
  • High radionuclidic purity.

This section has been updated according to the suggested edits.

  1. Table 1 should also include O-15 (used for PET evaluation of oxygen metabolism and blood flow) and N-13 (also for blood flow imaging)

Table 1 has now been updated to include N-13 and O-15.

Specific comments:

  1. Line 112 – “However, these fluorophores are …” – please check this phrase and reword for clarity
  2. Line 120 – “…is less likely to negatively influence the biological…small molecules.”
  3. Line 191 – replace ‘could not be visualized with the eyes alone’ with ‘could not be visualized with the naked eye’

Thank you for pointing out these details, they have been corrected.

Reviewer 2 Report

Over the last two decades the fusion of two or more imaging techniques has been considered as advanced approach in detecting and treatment of cancer. In particular, molecular imaging using PET has gained increasing importance for the noninvasive evaluation and characterization of primary neoplasms and metastases, for planning surgery, therapy monitoring, prognostication and other application. In its turn, fluorescence imaging has been demonstrated to be a superior method for intra operative tumor detection using different tumor-specific probes. PET/fluorescence dual modality imaging might greatly benefit the patient treatment because the lesion could be located using preoperative in-vivo PET scans, while the optical motif in the same probe molecule would allow surgeons to identify the PET-detected lesions or smaller metastasis in intra operative image-guided surgery. The purpose of this article is to summarize advances in PET fluorescence resolution, PET/fluorescence probes design using different scaffolds and PET radionuclides as well as the results of preclinical imaging. As for clinical PET fluorescence imaging, this area of application is on its initial stage and only one example is given in the review. In this context, it would be nice to cite and discuss the recent article by Aras O et al. on the “Small Molecule, Multimodal, [18F]-PET and Fluorescence Imaging Agent Targeting Prostate-Specific Membrane Antigen: First-in-Human Study”. Clin Genitourin Cancer. 2021 (5):405-416. doi: 10.1016/j.clgc.2021.03.011.

Otherwise, the submitted review is a perfect match and significantly advancing the field. It is very well written; the content and the style is clear und understandable.

            To conclude, the manuscript is recommended for the publication after minor corrections in accordance to reviewer comments.

Author Response

We thank the reviewer for the guidance and their valuable comments to improve the quality of our submission. Please find our responses below:

Over the last two decades the fusion of two or more imaging techniques has been considered as advanced approach in detecting and treatment of cancer. In particular, molecular imaging using PET has gained increasing importance for the noninvasive evaluation and characterization of primary neoplasms and metastases, for planning surgery, therapy monitoring, prognostication and other application. In its turn, fluorescence imaging has been demonstrated to be a superior method for intra operative tumor detection using different tumor-specific probes. PET/fluorescence dual modality imaging might greatly benefit the patient treatment because the lesion could be located using preoperative in-vivo PET scans, while the optical motif in the same probe molecule would allow surgeons to identify the PET-detected lesions or smaller metastasis in intra operative image-guided surgery. The purpose of this article is to summarize advances in PET fluorescence resolution, PET/fluorescence probes design using different scaffolds and PET radionuclides as well as the results of preclinical imaging. As for clinical PET fluorescence imaging, this area of application is on its initial stage and only one example is given in the review. In this context, it would be nice to cite and discuss the recent article by Aras O et al. on the “Small Molecule, Multimodal, [18F]-PET and Fluorescence Imaging Agent Targeting Prostate-Specific Membrane Antigen: First-in-Human Study”. Clin Genitourin Cancer. 2021 (5):405-416. doi: 10.1016/j.clgc.2021.03.011.

Otherwise, the submitted review is a perfect match and significantly advancing the field. It is very well written; the content and the style is clear und understandable.

            To conclude, the manuscript is recommended for the publication after minor corrections in accordance to reviewer comments.

We thank the reviewer for their time and their comments. We have updated the manuscript to include this interesting article by Aras et al.

Reviewer 3 Report

This review by R. Yuen et al. covers the synthesis and applications in cancer of dual PET-FI probes. First, they summarise the choice of an appropriate PET radionuclide followed by the selection of the fluorophore. Then, they review reported probes based on the type of scaffold; peptides, small molecules, antibodies and briefly, nanoparticles.

This short review is well organised and cover the most important examples reported up to date. I think it can be valuable and interesting for readers working on the field. However, some major changes should be performed before recommending its publication in Pharmaceutics.

Major issues

  • Although the review is fully devoted to molecular imaging in cancer using PET-FI probes, they authors do not show any imaging example in the figures.
  • Overall, I think the review is descriptive with a lack of criticism. Adding criticism over the described examples will add important information for the readers to decide which approach is more advantageous in a particular case.

Minor issues

  • In the selection of radionuclide section authors emphasised over the chemistry of 11C and 18 However, there is not description of the radiochemistries available for the labelling with the other radionuclides. They only introduce the concept of chelates with radiometals. I would extend a little bit the chemistry with radiometals since most of examples described in the manuscript use 68Ga and 64Cu.
  • Line 91: Define DOTA and NOTA
  • In figure 2, authors name the different compounds as 1a, 1b, 1c, etc. It is a bit confused since it is figure 2.
  • Naming compounds as Probe 2 or probe 3 is more confusing that clarifying.
  • Lines 265-266: new paragraph seems to be wrong there.
  • Line 394: Which Authors?

Author Response

We thank the reviewer for the guidance and their valuable comments to improve the quality of our submission. Please find our responses below:

This review by R. Yuen et al. covers the synthesis and applications in cancer of dual PET-FI probes. First, they summarise the choice of an appropriate PET radionuclide followed by the selection of the fluorophore. Then, they review reported probes based on the type of scaffold; peptides, small molecules, antibodies and briefly, nanoparticles.

This short review is well organised and cover the most important examples reported up to date. I think it can be valuable and interesting for readers working on the field. However, some major changes should be performed before recommending its publication in Pharmaceutics.

Major issues

  • Although the review is fully devoted to molecular imaging in cancer using PET-FI probes, they authors do not show any imaging example in the figures.

Thank you, and we agree. We have selected some images to be included throughout the manuscript.

  • Overall, I think the review is descriptive with a lack of criticism. Adding criticism over the described examples will add important information for the readers to decide which approach is more advantageous in a particular case.

Thank you, and we agree. We have taken some time to incorporate more comments and criticisms throughout the manuscript.

Minor issues

  • In the selection of radionuclide section authors emphasised over the chemistry of 11C and 18 However, there is not description of the radiochemistries available for the labelling with the other radionuclides. They only introduce the concept of chelates with radiometals. I would extend a little bit the chemistry with radiometals since most of examples described in the manuscript use 68Ga and 64Cu.

Thank you for mentioning this. The section on radiometals has been expanded.

  • Line 91: Define DOTA and NOTA

Thank you, this has been fixed.

  • In figure 2, authors name the different compounds as 1a, 1b, 1c, etc. It is a bit confused since it is figure 2.
  • Naming compounds as Probe 2 or probe 3 is more confusing that clarifying.

We were not too sure what naming scheme to use for compounds that did not have names, but in the end, we decided that using numbers in order of appearance would be the simplest. We also believe that since every compound is shown in a figure, this would not be too much of an issue.

  • Lines 265-266: new paragraph seems to be wrong there.

Thank you, this has been fixed.

  • Line 394: Which Authors?

Thank you, this has been fixed.

Round 2

Reviewer 1 Report

The authors have adequately addressed all comments raised by this reviewer. Thank you.

Reviewer 3 Report

The authors have addressed the comments accordingly and the manuscript has substantially improved. Therefore, I recommend the publication in its present form.